# Triple Synchronous Primary Malignant Tumors of the Liver, Kidney, and Lung in a Male Patient: Case Report and Systematic Review

**DOI:** 10.3390/diagnostics15243172

**Published:** 2025-12-12

**Authors:** Alexandru Vlad Oprița, Eduard Achim, Cornelia Nițipir, Nicolae Boleac, Alissia-Nicoleta Pilatec, Florin Andrei Grama

**Affiliations:** 1Department of Oncology, Faculty of Medicine, “Carol Davila” University of Medicine and Pharmacy, 050474 Bucharest, Romania; alexandruvladoprita@yahoo.com (A.V.O.); cornelia.nitipir@umfcd.ro (C.N.); 2Department of Medical Oncology, “Saint Nicholas” Hospital Pitești, 110124 Pitești, Romania; 3Faculty of Medicine, “Iuliu Hațieganu” University of Medicine and Pharmacy, 400012 Cluj-Napoca, Romania; pilatec.alissia.nicoleta@elearn.umfcluj.ro; 4Department of Medical Oncology, Agrippa Emergency Clinical Hospital, 011356 Bucharest, Romania; 5General and Esophageal Surgery Clinic, “Sf Maria” Clinical Hospital, 011192 Bucharest, Romania; nicolaeboleac@yahoo.com; 6Department of Surgery, “Carol Davila” University of Medicine and Pharmacy, 050474 Bucharest, Romania; florin.grama@umfcd.ro

**Keywords:** multiple primary malignant tumors, triple synchronous malignancies, liver cancer, renal cancer, lung cancer

## Abstract

**Background:** Triple primary malignant tumors (TPMTs) are extremely rare and represent a major diagnostic and therapeutic challenge. Their frequency has increased with advances in cancer detection and longer patient survival. **Case presentation:** We report the case of a 76-year-old male diagnosed with three synchronous primary malignancies involving the liver, left kidney, and right lung. Imaging revealed a hepatic mass with arterial enhancement and portal washout, a large left renal mass, and a cavitated pulmonary nodule. Histopathological and immunohistochemical evaluation confirmed three distinct tumors: well-differentiated hepatocellular carcinoma, chromophobe renal cell carcinoma, and invasive non-mucinous lung adenocarcinoma. A multidisciplinary oncology board recommended surgical resection of the liver and kidney lesions and stereotactic body radiotherapy for the lung tumor. The patient underwent hepatectomy and nephrectomy but experienced severe postoperative complications leading to multi-organ failure and death. **Results of the systematic review:** A systematic search identified 83 relevant cases of triple primary malignancies after full-text eligibility assessment. None of the 159 articles included after primary screening described a synchronous association of primary liver, kidney, and lung cancers. **Conclusions:** This case highlights the importance of thorough diagnostic assessment and individualized, multidisciplinary management in patients with multiple synchronous malignancies. To our knowledge, this is the first reported case of synchronous hepatocellular carcinoma, chromophobe renal cell carcinoma, and lung adenocarcinoma.

## 1. Introduction

Multiple primary malignant tumors (MPMTs) were first described by Billroth in 1889. The incidence of such cases has risen substantially due to improved diagnostic methods, longer life expectancy, and more effective cancer therapies [1,2,3].

According to the Surveillance, Epidemiology, and End Results (SEER) criteria, malignancies diagnosed within two months are considered synchronous, while those occurring after that period are metachronous [4]. The International Agency for Research on Cancer (IARC) defines multiple primaries as two or more histologically distinct malignancies that (1) are malignant by histology, (2) arise in different sites, (3) if in immediate vicinity, are separated by at least a 2 cm area of normal mucous tissue, or (4) arise in the same organ more than five years apart, with metastasis ruled out [5]. As per IARC-aligned definitions used in many clinical studies, synchronous malignancies are those diagnosed within 6 months of the first primary tumor.

The incidence of double primary cancers ranges from 0.59% to 3.7% [6], whereas triple primary malignancies are exceedingly uncommon, occurring in only 0.029–2.0% of cancer patients. Recent epidemiologic studies estimate that 11.7% of all cancer patients develop multiple primaries, with 0.5% representing triple cancers and fewer than 0.1% involving four or more [6,7,8].

We report a unique case of synchronous triple primary malignancies: hepatocellular carcinoma (HCC), chromophobe renal cell carcinoma (RCC), and invasive non-mucinous lung adenocarcinoma, an association not previously described in the literature. The case is accompanied by a systematic review of all previously reported synchronous triple malignancies to contextualize its rarity and clinical implications.

## 2. Case Presentation

A 76-year-old retired Caucasian male presented to our hospital in April 2024 with a two-month history of continuous pain in the right upper quadrant of the abdomen. His past medical history was significant for chronic obstructive pulmonary disease, hypertension, type II diabetes mellitus, obesity (body mass index 38 kg/m^2^), chronic hepatitis C virus infection. His surgical history included a laparoscopic cholecystectomy and lumbar disc hernia repair. The patient reported a long-standing history of cigarette smoking exceeding 100 pack-years and occasional alcohol consumption. He denied occupational exposure to toxins. His family history was notable for prostate cancer in his father.

In March 2024, a contrast-enhanced computed tomography (CT) scan of the thorax, abdomen, and pelvis revealed three distinct lesions: a large left renal mass measuring 9.5 × 7.5 × 8.5 cm, a right lower lobe pulmonary lesion measuring 2.8 × 2.6 × 4 cm, and a hepatic mass located in segments II and III measuring 10 × 10 × 9 cm. Additionally, a small tentorial meningioma was incidentally identified. It was deemed clinically insignificant and required no intervention. The hepatic lesion demonstrated intense enhancement during the arterial phase followed by portal venous washout, a pattern highly suggestive of primary hepatocellular carcinoma given the patient’s background of chronic hepatitis C infection.

In June 2024, an 18F-fluorodeoxyglucose positron emission tomography–computed tomography (FDG PET-CT) scan confirmed a centrally cavitated right lower lobe lesion with a maximum standardized uptake value (SUVmax) of 3.91, a left lobe hepatic lesion with mildly increased metabolic activity (SUV 5.22 compared to a background SUV of 4.28 in the rest of the hepatic parenchyma), and a left renal mass with slightly increased uptake (Figure 1). Serum tumor markers obtained in April 2024 were within normal limits: carcinoembryonic antigen (CEA) 3.34 ng/mL, carbohydrate antigen 19-9 (CA 19-9) 6.21 U/mL, and alpha-fetoprotein (AFP) 1.87 ng/mL.

**Figure 1 diagnostics-15-03172-f001:**
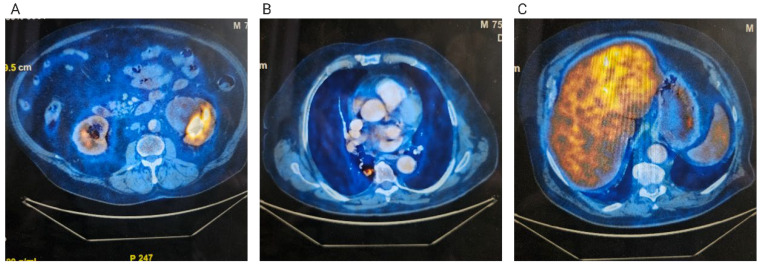
FDG PET-CT examination. (**A**) Left kidney tumor; (**B**) Right lung tumor; (**C**) Hepatic tumor.

On 27 May 2024, percutaneous biopsies of all three lesions were performed under combined CT and ultrasound guidance. Histopathological and immunohistochemical examination confirmed that each lesion represented a distinct primary malignancy (Figure 2, Table 1). The right lung tumor was positive for CK7 and thyroid transcription factor-1 (TTF-1), but negative for PAX8 and α-methylacyl-CoA racemase (AMACR). Clinical staging of the lung lesion was assessed according to the IASLC/AJCC TNM 9th edition (effective 1 January 2025). The lesion measured 4.0 cm in greatest dimension and was therefore staged as cT2cN0M0-Stage IB (IASLC TNM 9th edition). The left renal mass showed positivity for CK7, PAX8, AMACR, and CD117, with negative staining for CAIX, CD10, TTF1 and vimentin, supporting a diagnosis of chromophobe renal cell carcinoma (stage T2N0M0, stage II). The hepatic lesion was positive for arginase-1, heat shock protein 70 (HSP70), and glutamine synthetase, consistent with a well-differentiated hepatocellular carcinoma (stage T3N0M0, stage IIIA). Arginase-1 immunostaining was not available for the lung and kidney biopsy samples because the remaining tissue was insufficient for additional staining. However, the full IHC panels performed were adequate to confirm distinct primary origins.

Renal and liver cancer staging follows the AJCC TNM (8th edition), which was the version used in our institution at the time of diagnosis.

Per BCLC criteria, a solitary lesion >5 cm corresponds to BCLC stage B. This informed multidisciplinary decisions regarding resection. Preoperative Child–Pugh data and objective liver function tests were unavailable, which we note as a limitation.

Differential diagnosis included metastatic disease from a single primary. Distinct morphologic and immunophenotypic patterns excluded this possibility (Figure 2, Table 1). None shared a lineage marker pattern, and radiologic appearances were organ-specific. These findings support independent primaries.

**Figure 2 diagnostics-15-03172-f002:**
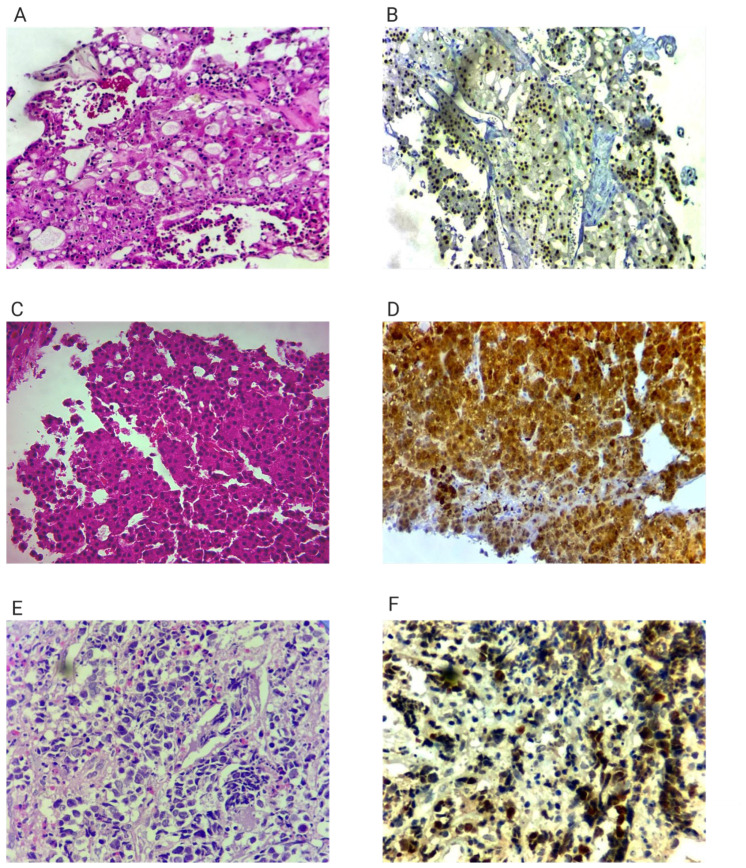
Histopathological and immunohistochemical features of the three tumors. Panels are grouped by tumor site to improve clarity, and all available essential diagnostic markers are presented. Because several original biopsy slides contained minimal remaining tissue, not all requested IHC panels could be retrieved or re-scanned. All available essential markers confirming independent primary origin are included.; (**A**) Left kidney tumor. Hematoxylin-Eosin Ob 20× (Zeiss Primostar 3, Oberkochen, Germany), solid and papilary features, polygonal cells with abundant eosinophilic cytoplasm and small nuclei without nucleoli; (**B**) Left kidney tumor—PAX 8 20× nuclear positivity in tumoral cells; (**C**) Hepatocellular carcinoma. Hematoxylin—Eosin 40× trabecular pattern with polygonal cells with nuclear atypia, including high N:C ratio, irregular nuclear membrane, multinucleation and prominent nuclei, cytoplasm varies from clear to eosinophilic; (**D**) Hepatocellular carcinoma. Glutamine synthetase 20× cytoplasmic and nuclear stain; (**E**) Right lung tumor. Hematoxylin—Eosin 40× pleomorphic nuclei with visible nucleoli and small amount of cytoplasm; (**F**) Right lung tumor. TTF1 40× nuclear positivity in tumoral nuclei.

**Table 1 diagnostics-15-03172-t001:** Histopathology and Immunohistochemistry findings (27 May 2024). Biopsies were obtained from all three lesions under CT and ultrasound guidance. TNM staging was determined using both preoperative imaging (clinical stage) and pathological findings from biopsy or surgical specimens, while immunohistochemical data were derived from the biopsy and surgical samples.

Tumor Site	Key Immunohistochemistry Findings	Diagnosis	TNM Stage
Right lung	CK7^+^, TTF-1^+^, PAX8^−^, AMACR^−^	Non-small cell lung carcinoma, invasive non-mucinous adenocarcinoma	cT2cN0M0 (IB)
Left kidney	CK7^+^, PAX8^+^, AMACR^+^, CD117^+^, CAIX^−^, CD10^−^, Vimentin^−^, TTF1^−^	Chromophobe renal cell carcinoma	T2N0M0 (II)
Liver	Arginase-1^+^, HSP70^+^, Glutamine synthetase^+^	Well-differentiated hepatocellular carcinoma	T3N0M0 (IIIA)

Following multidisciplinary discussion, the tumor board concluded that all three malignancies were localized and lacked a common systemic treatment option. Therefore, curative-intent surgical resection of the hepatic and renal tumors was recommended, with stereotactic body radiotherapy (SBRT) proposed for the pulmonary lesion, due to early-stage disease and the patient’s high surgical risk related to severe chronic lung dysfunction.

In August 2024, the patient underwent a left lateral hepatic sectionectomy followed by a left nephrectomy with adrenalectomy. Postoperative histopathological examination confirmed a moderately differentiated hepatocellular carcinoma (G2), staged as pT2 with an R1 resection, consistent with the preoperative diagnosis. The nephrectomy specimen demonstrated a renal carcinoma with oncocytic morphology, compatible with chromophobe renal cell carcinoma. Because this case was managed retrospectively and the full synoptic pathology reports were not accessible in the medical archive, additional details such as exact tumor dimensions, vascular invasion, and margin measurements were unavailable. All verified pathological information has been incorporated into the manuscript, and this limitation is acknowledged. The postoperative course was severely complicated. On the second postoperative day, the patient developed significant postoperative cognitive dysfunction. By the third day, a biliary fistula was identified and managed conservatively. On the 5th postoperative day, a CT scan with IV contrast was performed, which ruled out the presence of an intraperitoneal collection, except for a minimal accumulation with a thickness of 1 cm at the hepatectomy transection site. On the seventh postoperative day, he developed pancreatic suppuration with persistent biliary and peripancreatic drainage. On the tenth day, postoperative wound dehiscence with exteriorization of small bowel loops occurred, requiring emergency surgical intervention. During this reoperation, extensive lavage, suturing of biliary fistulas, drainage placement, and abdominal wall repair with mesh reinforcement were performed. On the ninth day post-reintervention, the patient undergoes a CT scan with IV contrast: results are comparable to the previous examination, except for the recurrence of the postoperative median hernia. On the tenth day post-reintervention, the patient presents with the exteriorization of a significant amount of serohematic fluid at the wound site; an emergency surgical reintervention is decided, during which an evisceration repair with PPMF mesh is performed.

Despite aggressive surgical and intensive care management, the patient’s postoperative evolution was marked by progressive neurological deterioration, acute renal failure requiring replacement therapy, and hematologic, cardiovascular, and respiratory dysfunction. On 25 September 2024, while under mechanical ventilation and vasopressor support, he experienced cardiac arrest due to asystole. Despite resuscitation efforts, spontaneous cardiac activity could not be restored, and death was pronounced at 10:25 a.m (Figure 3). Although hereditary cancer predisposition was considered, rapid clinical decline precluded formal germline testing.

**Figure 3 diagnostics-15-03172-f003:**
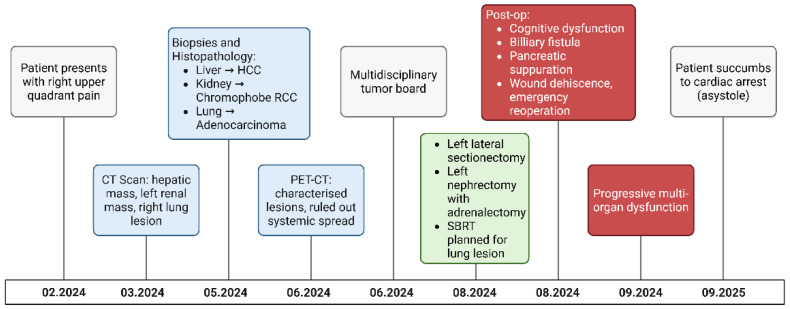
Patient timeline.

## 3. Systematic Review

The systematic search was conducted in PubMed, EMBASE, and Scopus from 24 August 2024 to 11 September 2025, using the following terms (combined with appropriate database syntax): (“multiple primary” OR “multiple primaries” OR “multiple primary malignant tumor” OR “multiple primary malignancies” OR “triple primary” OR “triple synchronous” OR “synchronous triple”) AND (cancer OR carcinoma OR neoplasm). The time interval of screened articles was not restricted. Article screening and eligibility assessment were performed independently by two reviewers (E.A., A.N.), with disagreements resolved by a third senior reviewer (A.O.). Reporting follows PRISMA guidelines, and a detailed flow diagram summarizing the literature search and selection process is shown in Figure 4. A protocol was not prepared.

The search yielded 483 results. After title and abstract screening, 159 articles describing triple synchronous primaries were retained. Secondary full-text screening applied established multiple-primary tumor criteria, including: histologic confirmation of malignancy, diagnosis of all tumors within 2 months, and clear evidence of distinct primary origins. Reports containing benign tumors, prolonged diagnostic intervals, or lacking adequate pathological confirmation were excluded. A total of 83 articles met all criteria and were included in the final review.

Data were extracted independently by both reviewers using a standardized form and are presented in tabular form in Appendix A. For each included case we extracted the year of publication, patient details, diagnostic investigations, histology, tumor sites, treatment and survival outcomes. Because all eligible studies were case reports or small series, formal risk-of-bias assessment was not applicable; however, potential publication bias was recognized as a limitation. Because of extreme clinical heterogeneity and likely publication bias, no quantitative pooling or statistical summary was performed. The PRISMA 2020 checklist is provided in Appendix A, and this review was not registered in PROSPERO or any other database. CARE checklist is available in Appendix A.

**Figure 4 diagnostics-15-03172-f004:**
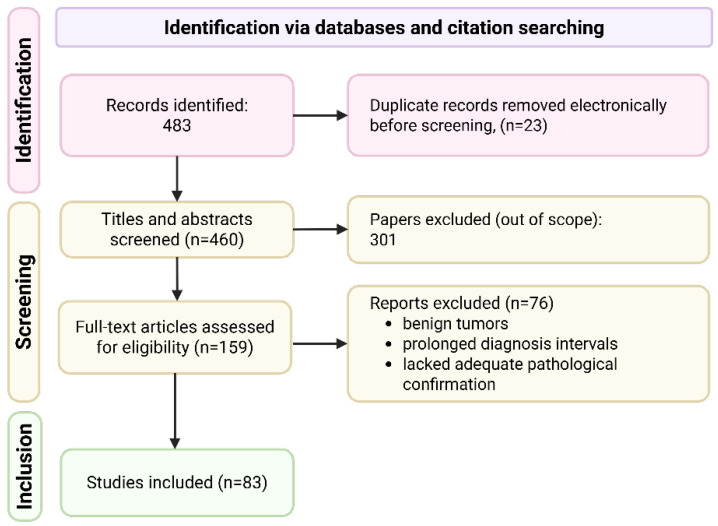
PRISMA flow diagram of literature search and study selection. The systematic search identified 483 records across PubMed, Embase, and Scopus. After removal of 23 duplicates, 460 unique records were screened by title and abstract, and 301 were excluded as out of scope. A total of 159 full-text articles were assessed for eligibility; 76 were excluded (benign tumors, prolonged diagnostic intervals, or inadequate pathological confirmation). Finally, 83 studies met all inclusion criteria and were included in the qualitative synthesis (Appendix A).

Eligible cases were categorized by organ-system involvement: digestive tract [9,10,11,12,13,14,15,16,17,18,19,20,21,22,23,24,25,26,27,28], respiratory tract [29,30,31,32,33,34,35,36], genitourinary tract [37,38,39,40,41,42,43,44,45,46,47,48,49,50], head and neck [51,52,53,54,55,56,57], skin [58], and mixed-system combinations [59,60,61,62,63,64,65,66,67,68,69,70,71,72,73,74,75,76,77,78,79,80,81,82,83,84,85,86,87,88,89,90]. Digestive malignancies encompassed cancers of the esophagus, stomach, small intestine, colon, appendix, and hepatopancreatobiliary system. Genitourinary cases included kidney, urinary bladder, prostate, and both male and female reproductive organs. Respiratory cases consisted almost exclusively of independent primary lung tumors, with one case including synchronous laryngeal disease. Lymphomas confined to a single organ were classified by that organ; disseminated lymphomas were categorized within the lymphatic system.

Notably, none of the 159 screened reports documented simultaneous primary malignancies of the liver, kidney, and lung, underscoring the exceptional rarity of the presentation described in our case.

Across the compiled series the single largest category was mixed/different-system combinations (32 cases), followed in descending order by digestive (21 cases), genitourinary (14 cases), respiratory (8 cases), head and neck (7 cases) and a solitary cutaneous entry (1 case) (Table 2).

**Table 2 diagnostics-15-03172-t002:** Summary of triple synchronous malignancies reported in the literature according to organ system involvement. Detailed individual case data are provided in Appendix A.

Organ System	Number of Cases	Representative Sites	Percentage of Total (%)
Mixed organs/different systems	32	Various organ combinations	38.55
Digestive system	21	Stomach, colon, liver, pancreas	25.30
Genitourinary system	14	Kidney, bladder, prostate	16.87
Respiratory tract	8	Lung, larynx	9.64
Head and neck	7	Oral cavity, thyroid, pharynx	8.43
Skin	1	Melanoma	1.21
Total	83	-	100

This distribution shapes several modest, evidence-anchored observations. Mixed-system combinations are not random curiosities but a recurring pattern in the dataset: many reports pair anatomically and embryologically unrelated primaries (for example breast–colon–kidney, lung–thyroid–bladder, or ovary–kidney–lung), suggesting that when three primaries occur synchronously they often reflect systemic predisposition (genetic susceptibility, widespread carcinogenic exposure, or age-related genomic instability) rather than a single local field effect. Representative mixed-system cases in the table include breast–rectum–bladder and colon–kidney–lung triads, among others.

Digestive-tract triads form the next largest cluster and typically involve combinations of colorectal, gastric, pancreatic or hepatobiliary primaries. These groupings likely reflect a mix of shared environmental risk factors (diet, alcohol, chronic viral liver disease), organ proximity with synchronous detection during staging or laparotomy, and the high baseline incidence of gastrointestinal cancers; several entries document multiple gastrointestinal tumors discovered at the same operation or on the same imaging workup.

Genitourinary combinations often center on renal primaries occurring alongside bladder, prostate or gynecologic tumors; the dataset contains multiple examples of renal–urothelial–prostatic and ovarian–endometrial–tubal triads. Shared risk exposures (smoking, occupational toxins), hormonal influences in pelvic organs, and the use of cross-sectional pelvic imaging that concurrently visualizes multiple genitourinary structures probably contribute to this pattern.

Respiratory and head-and-neck triads are frequently populated by squamous histologies and by multiple independent lung primaries; smoking and field cancerization of the aerodigestive tract remain plausible unifying mechanisms for several of these reports. The single cutaneous case (three independent melanomas) is consistent with the known tendency of melanoma patients to develop multiple primary cutaneous lesions but remains an outlier in the overall series.

In sum, the predominance of mixed-system presentations followed by digestive and genitourinary clusters argues that synchronous triple primaries are most often a manifestation of systemic susceptibility or shared exposures, with an important role for diagnostic cross-sectional imaging and intraoperative discovery in shaping which combinations are reported.

Adenocarcinoma emerged as the dominant histologic lineage across most organ groups, consistent with its high baseline incidence in gastrointestinal, pulmonary, and genitourinary malignancies. Squamous cell carcinomas were the next most frequent, particularly in lung and head-and-neck triads. Neuroendocrine tumors appeared sporadically across systems, and lymphomas were occasionally included when clearly organ-confined. Rare entities such as sarcomatoid carcinomas, granulosa cell tumor, thymoma, and stromal tumors were also represented, underscoring that triple synchronous disease is not restricted to a single biologic pathway. Only a minority of the most recent case reports incorporated molecular testing such as NGS or targeted mutation analysis; most relied on traditional histopathology and immunohistochemistry to define tumor independence.

Treatment approaches are dominated by surgical management: colectomies, gastrectomies, pancreatectomies, hepatectomies, nephrectomies, lung resections (wedge resection, lobectomy, pneumonectomy), and radical procedures (cystoprostatectomy, hysterectomy) are repeatedly reported. Many reports pair surgery with adjuvant or neoadjuvant therapies, including chemotherapy regimens (5-FU, cisplatin combinations, capecitabine/oxaliplatin, platinum-based doublets), radiotherapy, or locoregional procedures (transarterial embolization, microwave ablation). A minority of cases document palliative management or decision to forgo active oncologic therapy. These therapeutic patterns indicate that when triple synchronous tumors are anatomically resectable, combined surgical strategies are commonly pursued, sometimes in combination with systemic or local adjuvant treatments.

Demographically, reported ages ranged from the early 30s to over 80 years, with the expected enrichment in older adults who carry a higher cumulative risk for multiple malignancies. Both sexes were represented, with sex distribution generally mirroring the epidemiology of the organs involved; for example, gynecologic triads occurred exclusively in women, while several renal-urothelial-prostatic combinations occurred in men. Follow-up data were inconsistently documented: some cases reported long-term survival exceeding several years, whereas others described early postoperative mortality or treatment-related complications, reflecting the heterogeneity in tumor aggressiveness, comorbidity, and treatment feasibility across this rare clinical entity.

In summary, the systematic search indicates that triple synchronous malignancies present most commonly within the gastrointestinal, genitourinary and respiratory domains and that the dominant management strategy recorded is surgical resection often combined with site-specific adjuvant therapy. The histologic variety underlines that triple synchronous disease is not restricted to a single tumor type and that individual case management is guided mainly by anatomical resectability and organ-specific standards of care. For transparency and reproducibility, a detailed description of all identified cases, including patient characteristics, tumor histology and treatment approach is provided in Appendix A.

These claims are grounded in the case patterns and representative examples tabulated in Appendix A, and require cautious interpretation, since the available literature consists almost entirely of isolated case reports and small case series rather than systematic, population-level registries. Such reports are influenced by publication bias, frequently highlighting unusual presentations while underreporting more routine combinations. Diagnostic and reporting practices vary widely across decades and institutions, meaning that some tumors may have gone undetected, misclassified, or judged to be metastatic rather than independent primaries. In several older studies, pathology confirmation was limited, and comprehensive genomic or molecular profiling was rarely performed.

Consequently, the distributions described here likely reflect what has been documented, not necessarily the true epidemiology of synchronous triple primaries in clinical practice. The findings may therefore be understood as signals drawn from the published record rather than definitive statements about incidence, causation, or risk patterns.

## 4. Discussion

Synchronous triple primary malignancies are extraordinarily rare clinical entities, often representing the intersection of cumulative carcinogenic exposures, underlying host susceptibility, and stochastic mutational events. To our knowledge, and according to a systematic review of PubMed, Embase, and Scopus through 11 September 2025, no prior reports described synchronous hepatocellular carcinoma, chromophobe renal cell carcinoma, and lung adenocarcinoma in a single patient.

Triple synchronous primaries impose a significantly greater physiological and therapeutic burden compared with double primaries. Prognosis depends less on any single tumor and more on the combined operative risk, treatment feasibility, and overall functional reserve of the patient [91].

Diagnostic certainty is essential, since misclassification of metastasis as a primary tumor may lead to overtreatment [92,93]. In the absence of universal molecular profiling, organ-specific histopathology and immunohistochemistry remain the foundation of determining tumor independence. As sequencing becomes more accessible, clonality assessment may assume a greater role in future diagnostic algorithms. In practice, management prioritizes the tumor that poses the most immediate threat to life or organ function, while avoiding excessive surgical trauma in patients with limited reserve. Sequential or minimally invasive strategies may improve outcomes.

Recent molecular studies support the role of germline variants (e.g., TP53, BRCA2, PTEN, and mismatch repair genes) and epigenetic deregulation in predisposing to multiple primaries [94]. While next-generation sequencing (NGS) was not performed in this patient, it could have clarified whether a common genomic predisposition existed or whether the three tumors arose through independent mutational events. Such analyses are increasingly relevant for precision oncology and familial risk assessment.

Emerging recommendations support routine evaluation for hereditary predisposition, especially in younger patients or those with tumor types known to associate with germline risk. Genetic assessment may enable tailored surveillance for second and third primary tumors and facilitate counseling for family members. Future multi-institutional registry efforts will be crucial to clarify optimal treatment sequencing and to better estimate true prognostic outcomes.

Management of MPMTs must be individualized and multidisciplinary. When feasible, surgical resection of localized lesions remains the cornerstone of therapy. However, as illustrated here, the cumulative physiological stress of extensive surgery can lead to high postoperative morbidity, especially in elderly or comorbid individuals. In selected patients, minimally invasive or sequential therapeutic strategies, including ablative or radiotherapeutic approaches, may reduce treatment-related risk while maintaining disease control.

The present case illustrates the interplay between established risk factors, such as age, tobacco use, obesity, and chronic hepatitis C, in driving independent tumorigenic pathways across different organs.

In men with renal cell carcinoma, the lifetime risk of a second primary malignancy may reach 26.6% [95]. The association of chromophobe RCC with other solid tumors has been noted, but synchronous presentation with hepatocellular carcinoma and lung adenocarcinoma is exceptional. Shared etiologic contributors such as oxidative stress, chronic inflammation, and impaired DNA repair mechanisms may create a “field effect” promoting neoplastic transformation in multiple tissues.

This case also underscores the importance of differentiating multiple primaries from metastases, as management and prognosis differ radically. Immunohistochemistry and molecular profiling remain indispensable for establishing clonality and guiding treatment. This report is strengthened by comprehensive pathologic confirmation of three independent primaries and an accompanying systematic literature review.

Quantitative summarization of the identified reports was intentionally avoided, as the available literature is subject to substantial publication bias favoring unusual or successful cases. The systematic review was therefore designed to document completeness and novelty rather than estimate incidence or outcomes.

Limitations include the anecdotal nature of single-case observations and inherent publication bias in available evidence, which restricts broader generalization. Next-generation sequencing or germline testing could not be performed because of tissue unavailability, precluding molecular confirmation of independence among tumors. Another limitation of this report is the unavailability of arginase-1 staining for the pulmonary and renal tumors due to limited residual biopsy material. These findings should therefore be interpreted as reflective of published experience rather than true population-level incidence.

## 5. Conclusions

Triple synchronous primary malignancies are among the rarest oncological phenomena. This case emphasizes the necessity of comprehensive diagnostic work-up, meticulous histopathologic evaluation, and multidisciplinary decision-making in suspected cases. As diagnostic precision and patient survival continue to improve, such complex presentations are likely to become more frequently recognized.

To our knowledge, this is the first reported case of synchronous hepatocellular carcinoma, chromophobe renal cell carcinoma, and lung adenocarcinoma in a single patient: a reminder of the multifactorial nature of carcinogenesis and the ongoing need for individualized oncologic strategies.

## Data Availability

The original contributions presented in this study are included in the article/Appendix A. Further inquiries can be directed to the corresponding author.

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
