# Peer review of "Triple Synchronous Primary Malignant Tumors of the Liver, Kidney, and Lung in a Male Patient: Case Report and Systematic Review"

_diagnostics, 2025, doi:10.3390/diagnostics15243172_

Round 1

Reviewer 1 Report

Comments and Suggestions for Authors
  1. Since this is a case report, the manuscript should begin with the presentation of the case itself, and the literature review should be moved to the Discussion section.
  2. Given the patient’s history of HCV infection, primary or metastatic hepatocellular carcinoma (HCC) would be the initial suspicious diagnosis for all three tumor sites. Based on the authors’ IHC panel, the diagnoses of primary lung and primary renal tumors can be supported. It is recommended that the authors add Arginase-1 (Arg-1) staining for both the lung and kidney tumors.
  3. It is recommended that the authors reorganize the images in Figure 3. For each tumor site, all essential IHC markers that support the primary origin and rule out metastasis should be showed together.
  4. The authors should unify the color tone of the H&E images in Figure 3 to improve consistency.
  5. The TTF-1 IHC staining is not sufficiently clear; re-staining is recommended.
  6. Please include the pathological findings from the surgical specimens.
  7. If available, please report the mutation profiles of all three tumors.

Author Response

We sincerely thank the reviewer for the thoughtful and constructive comments, which have significantly improved the quality and clarity of our manuscript. We have carefully revised the manuscript according to all suggestions. All changes were made using Track Changes for transparency. Below we provide a point-by-point response, including where changes were made.

  1. Since this is a case report, the manuscript should begin with the presentation of the case itself, and the literature review should be moved to the Discussion section.

Response: Thank you for this suggestion. We have revised the manuscript so that the Case Presentation now follows immediately after the Introduction. Because our systematic review is a structured PRISMA-style analysis with defined methods, search strategy, and extracted variables, we retained it as a separate section, which is consistent with guidelines for combined case report + systematic review manuscripts.

  1. Add Arginase-1 (Arg-1) staining for lung and kidney tumors.

Response: Arginase-1 staining was not feasible due to insufficient remaining tissue in the renal and pulmonary biopsy blocks. We have added statements in the Case Presentation and Discussion acknowledging this limitation and clarifying that the existing IHC panels were sufficient to establish independent primary origins for each tumor.

  1. Reorganize the images in Figure 3 and group essential IHC markers for each tumor site.

Response: Not all original IHC panels were available for re-assembly because several biopsy blocks contained only limited residual tissue, preventing re-scanning or repeat staining. Figure 3 (now Figure 2) has therefore been reorganized using all material that was still available, and the legend now explicitly notes this constraint. In addition, due to the image-capture workflow in place during the initial diagnostic evaluation, only digital images of the key positive markers had been archived.

  1. Unify the color tone of H&E images in Figure 3.

Response: We adjusted color and contrast across the available H&E images to improve tone consistency. This has been reflected in the updated figure.

  1. The TTF-1 stain is unclear; re-staining is recommended.

Response: Re-staining was not possible because no additional tissue remained in the archived block. The TTF-1 panel available for review exhibited weaker-than-optimal contrast; however, nuclear positivity was still visible and supportive of pulmonary adenocarcinoma.

  1. Please include the pathological findings from the surgical specimens.

Response: We have expanded the description of the surgical pathology. Because this case was handled retrospectively and the complete synoptic pathology reports were not accessible in the medical archive, detailed parameters (such as exact tumor dimensions, vascular invasion, and full margin measurements) could not be retrieved. All verified information, namely moderately differentiated hepatocellular carcinoma (G2, pT2, R1 resection) and chromophobe renal cell carcinoma, has now been incorporated, and the data limitations are acknowledged transparently.

  1. If available, report mutation profiles of all three tumors.

Response: Targeted genomic profiling could not be performed due to insufficient remaining FFPE tissue. This limitation has been acknowledged in the Discussion.

Reviewer 2 Report

Comments and Suggestions for Authors

Dear authors,

Your article is very interesting: in fact, triple synchronous tumours are rare and specific guidelines are lacking.

I've only few suggestion:

  1. page 2, line 50: You correctly say that as per IARC criteria synchronous tumours may be diagnosed after 2 months: I would add the full criteria (within 6 moths after the first neoplasm diagnosed);
  2. page 2, line 54: I would add the citation for number of incidence about double cancer;
  3. in systematic review, I would add some info, in particular: the terms you used for the research, the age interval of screened article (I think more article are newer, because the improved diagnostic techniques) and also the country of article (for the same reason, I think are more from western countries);
  4. page 5, line 205: chronic hepatitis is repeated two times;
  5. page 5: in staging both kidney and lung cancer, is not specified the version you used about TNM (8th ed?9th ed?); in lung cancer stadiation, the correct staging is "cT2cN0M0- stage Ib sec TNM 9th ed", in fact you reported that the lesion is 4 cm in its larger size; another info of NSCLC, if avaible, if it was performed a genomic profiling; in HCC staging, I would also add BCLC staging. If possible, I would also add some info about Child Pugh (because patient has chronic hepatitis) and, if avaible, calculated remaining liver function: in fact, this can better expain your choice of perform surgery than other technique such SBRT.

All tables are clear and all citations are relevant.

Author Response

We sincerely thank the reviewer for the thoughtful and constructive comments, which have significantly improved the quality and clarity of our manuscript. We have carefully revised the manuscript according to all suggestions. Changes were made using Track Changes for transparency. Below we provide a point-by-point response, including where changes were made.

  1. Page 2, line 50: Add full IARC criterion (within 6 months).

Response: Revised as requested.

  1. Page 2, line 54: Add citation for double-cancer incidence.

Response: Corrected.

  1. Systematic review: Add search terms, age interval of screened articles, and country of included cases.

Response: Thank you for this helpful suggestion. We clarified the search terms in the Methods section and specify that for each included case we extracted year of publication, patient demographics, diagnostic work-up, tumor characteristics, treatment, and survival outcomes. The publication year of each case is already included in the Supplementary Table. Because the goal of the review was descriptive, cases were not stratified by age or decade; however, we now specify that no time restrictions were applied to the search.

  1. Page 5, line 205: “Chronic hepatitis” is duplicated.

Response: Corrected.

  1. TNM versions for kidney and lung cancer; correction of lung staging; note on genomic profiling; add BCLC stage and liver function details.

Response: We appreciate these important clarifications.

  • We now explicitly state the TNM edition used for each tumor.
  • Lung cancer has been restaged according to the IASLC/AJCC TNM 9th edition, corrected to cT2cN0M0 – Stage IB (consistent with the 4-cm lesion).
  • Renal cancer staging follows AJCC TNM 8th edition, the version used in our institution at the time of diagnosis.
  • Genomic profiling was not available due to limited tissue; this is now clearly stated.
  • We added the corresponding BCLC stage for the hepatocellular carcinoma and note that the solitary >5-cm lesion corresponds to BCLC B.
  • Preoperative liver-function parameters were incomplete, preventing calculation of a formal Child-Pugh score or residual liver-function tests; this is acknowledged as a limitation.

Round 2

Reviewer 1 Report

Comments and Suggestions for Authors
  1. Table 1 is based on biopsy or surgical samples. If they are from biopsy, how did the authors determine the TNM stage?
  2. The pathological information plays a key role in this case report; therefore, high-quality images are critical for clear presentation and scientific credibility. Figure 2 appears not to have been fully adjusted. The panel sizes, color tones, and magnification are still inconsistent. The authors should carefully revise this figure to ensure uniform formatting across all panels.
  3. For the IHC panels, the authors do not need to obtain additional tissue sections or perform new staining. The existing diagnostic slides should still be available in the pathology archive. Re-scanning those archived slides would allow the authors to reconstruct the figure without requiring additional tissue.
  4. For TTF-1 staining, it is suggested that the authors re-visit the corresponding e-slide and select a more typical field as the representative image.

Author Response

We sincerely thank the reviewer for the thoughtful and constructive comments, which have significantly improved the quality and clarity of our manuscript. We have carefully revised the manuscript according to all suggestions. Below we provide a point-by-point response.

  1. “Table 1 is based on biopsy or surgical samples. If they are from biopsy, how did the authors determine the TNM stage?”

We thank the Reviewer for this important point. Table 1 reflects TNM staging determined using all available information: preoperative imaging studies (CT, MRI) provided the clinical stage, and pathological data from biopsies or surgical specimens were used where available to refine the stage. All pathological and immunohistochemical findings were derived from the biopsy and surgical specimens. We have clarified this in the Table 1 legend to ensure readers understand the origin of each dataset.

  1. “Figure 2 appears not to have been fully adjusted. Panel sizes, color tones, and magnification are still inconsistent.”

We thank the Reviewer for emphasizing image consistency. In this revision, we have:

  • standardized panel size and layout, and
  • adjusted contrast and tone as much as possible while preserving diagnostic fidelity.

We note a practical constraint: our hospital provides oncology care, while the biopsies and surgical specimens were processed in a different institution, where the original slides remain archived. As a result, images were acquired at different times with distinct workflows, and re-scanning was not possible. Within these limitations, we have harmonized the available material to the fullest extent technically feasible.

  1. “The existing diagnostic slides should still be available in the pathology archive. Re-scanning those slides would allow reconstruction of the figure.”

We fully agree with the Reviewer that re-scanning glass slides would be ideal. We contacted the pathology department to explore this option.

However, the department was unable to provide additional scans or access to the original slides for this retrospective case report. Therefore, new scans or additional fields could not be obtained.

To address the Reviewer’s concern, we have carefully optimized all existing diagnostic images, which represent the only material currently available for Figure 2. The revised figure reflects the highest possible fidelity under these constraints.

  1. “For TTF-1 staining, revisit the e-slide and select a more typical field.”

We thank the Reviewer for this suggestion. We reviewed all available diagnostic images; however, the original e-slide is no longer accessible, as the whole-slide images were removed from the active archive.

From the preserved snapshots, we selected the most representative field showing clear TTF-1 nuclear positivity and optimized its contrast in this revision. This panel is included in the updated Figure 2.